# l-Ornithine-l-Aspartate (LOLA) Normalizes Metabolic Parameters in Models of Steatosis, Insulin Resistance and Metabolic Syndrome

**DOI:** 10.3390/pharmaceutics16040506

**Published:** 2024-04-07

**Authors:** Ali Canbay, Oliver Götze, Ozlem Kucukoglu, Sönke Weinert, Roland S. Croner, Theodor Baars, Mustafa K. Özçürümez, Robert K. Gieseler

**Affiliations:** 1Department of Internal Medicine, University Hospital, Knappschaftskrankenhaus, Ruhr University Bochum, 44892 Bochum, Germanytheodor.baars@t-online.de (T.B.); mustafa.oezcueruemez@kk-bochum.de (M.K.Ö.); 2Department of Gastroenterology, Hepatology and Infectious Diseases, Medical Faculty, Otto-von-Guericke University, 39120 Magdeburg, Germany; kucukoglu.ozlem@gmail.com; 3Department of Cardiology, Medical Faculty, Otto-von-Guericke University Magdeburg, 39120 Magdeburg, Germany; soenke.weinert@med.ovgu.de; 4Department of General, Visceral, Vascular and Transplantation Surgery, Medical Faculty, Otto-von-Guericke University, 39120 Magdeburg, Germany; roland.croner@med.ovgu.de

**Keywords:** NAFL, NASH, ammonia, steatosis, insulin resistance, metabolic syndrome, mitochondria

## Abstract

l-Ornithine- l-aspartate (LOLA) reduces toxic ammonium (NH_3_) plasma levels in hepatic encephalopathy. NH_3_ detoxification/excretion is achieved by its incorporation into urea and glutamine via activation of carbamoyl phosphate synthetase 1 (CSP1) by l-ornithine and stimulation of arginase by l-aspartate. We aimed at identifying additional molecular targets of LOLA as a potential treatment option for non-alcoholic fatty liver disease (NAFLD). In primary hepatocytes from NAFLD patients, urea cycle enzymes CSP1 and ornithine transcarbamylase (OTC) increase, while the catabolism of branched-chain amino acids (BCAAs) decreases with disease severity. In contrast, LOLA increased the expression rates of the BCAA enzyme transcripts *bcat2*, *bckdha*, and *bckdk*. In untreated HepG2 hepatoblastoma cells and HepG2-based models of steatosis, insulin resistance, and metabolic syndrome (the latter for the first time established herein), LOLA reduced the release of NH_3_; beneficially modulated the expression of genes related to fatty acid import/transport (*cd36*, *cpt1*), synthesis (*fasn*, *scd1*, ACC1), and regulation (*srbf1*); reduced cellular ATP and acetyl-CoA; and favorably modulated the expression of master regulators/genes of energy balance/mitochondrial biogenesis (AMPK-α, *pgc1α*). Moreover, LOLA reconstituted the depolarized mitochondrial membrane potential, while retaining mitochondrial integrity and avoiding induction of superoxide production. Most effects were concentration-dependent at ≤40 mM LOLA. We demonstrate for l-ornithine-l-aspartate a broad range of reconstituting effects on metabolic carriers and targets of catabolism/energy metabolism impaired in NAFLD. These findings strongly advocate further investigations to establish LOLA as a safe, efficacious, and cost-effective basic medication for preventing and/or alleviating NAFLD.

## 1. Introduction

The amino acids l-ornithine and l-aspartate play key roles in the detoxification of ammonia (NH_3_) and in the biosynthesis of proline and polyamines. The latter are considered critical in the context of DNA synthesis, cell replication, and hepatic regeneration. In health, l-ornithine and l-aspartate are synthesized de novo in sufficient quantities. However, in conditions such as disease—including non-alcoholic liver disease (NAFLD) and urea cycle disorders—tissue damage, organ insufficiency, excessive metabolic demand, growth, or pregnancy, they have to be supplied by diet [1].

The stable salt of l-ornithine and l-aspartic acid—l-ornithine-l-aspartate (LOLA)—has been demonstrated to reduce the NH_3_ plasma concentration and improve the psychometric performance in patients with hyperammonemia and hepatic encephalopathy (HE) due to acute liver failure [2,3,4,5]. Supposed mechanisms of action of LOLA are the stimulation of the urea cycle and of glutamine synthesis, which are assumed to enhance NH_3_ detoxification [3,4,5,6]. LOLA also ameliorated the deleterious psychometric effects of glutamine in Child’s grade B and C patients with cirrhosis without transjugular intrahepatic portosystemic shunts [7]. Additional findings were complete normalization of serum γ-glutamyltransferase upon LOLA treatment in patients with chronic liver disease [8] and reduced times of hospitalization in patients with liver cirrhosis and HE [9]. Meanwhile, increased concentrations of NH_3_ have also been detected in the livers of patients with NAFLD [10]. At this point, we need to mention that an international consensus panel early in 2020 suggested the term ‘metabolic (dysfunction) associated fatty liver disease’ (or MAFLD) as a more appropriate term for defining the condition hitherto referred to as NAFLD [11]. In this present article, we adhere to the established nomenclature. In the future, however, it may indeed become useful to implement the authors’ proposed new definition, as this could, according to their reasoned opinion, accelerate the path to new therapies as a result of a scrupulous re-examination and updating of the overall nomenclature and subphenotypes of this disease [11].

In recent years, molecular mechanisms of LOLA have been investigated in greater detail. Once administered, LOLA readily dissociates into its constituent amino acids. In the urea cycle in periportal hepatocytes, l-ornithine serves as an intermediary and activates carbamoyl phosphate synthetase 1, and l-aspartate stimulates the activity of arginase. As a result, NH_3_ is incorporated into urea and glutamine, thus confirming LOLA as an effective NH_3_-lowering treatment in HE [12,13]. Meta-analyses of randomized controlled trials conducted over the past two decades evidenced the benefit of LOLA for the mental state in both overt and minimal HE; the oral LOLA formulation was particularly effective [14]. The potential therapeutic value of LOLA appears to extend well beyond its use in chronic hepatitis C [15]. However, it does not appear to have any benefit in acute liver failure [16].

As NAFLD is the leading and steadily increasing cause of chronic liver disease worldwide, and since treatment options are limited, we recently highlighted the possibility of harnessing LOLA’s hepatoprotective properties especially for this disease entity [17].

In the present work, we investigated whether the expression of putative targets of LOLA is altered in NAFLD patients in a manner that would suggest a therapeutic use of LOLA in this complex disorder. In parallel, a broad range of hepatocellular functions and mechanisms related to steatosis, insulin resistance and the metabolic syndrome (MetS) was assessed in suitable in vitro models of metabolic injury. We found that, in addition to the known effects of LOLA on NH_3_ detoxification, it obviously normalizes fatty acid transport regulation, branched-chain amino acid (BCAA) catabolism, energy consumption, and the mitochondrial energy balance. Thus, it appears that LOLA-dependent effects on the parameters studied here may be desirable for improving the management of NAFLD. However, the results of this exploratory study will have to be confirmed using larger sample sizes. We have reserved the obvious value of comparing the effects of LOLA with its singular components as well as with representatives of other amino acid classes for a follow-up study (see also *Discussion*).

## 2. Materials and Methods

### 2.1. Patients

*(i) Ethics:* The study protocol conformed to the revised 2008 Declaration of Helsinki. The protocols were approved by the Ethics Committees of (i) the Medical Faculty of the University Duisburg-Essen, Germany, for morbidly obese NAFLD patients undergoing bariatric surgery (BAS) (file number: 09-4252, and (ii) the Medical Faculty of Otto-von-Guericke University, Germany, for patients undergoing liver resection due to hepatic metastases (file number: 208/17). All patients provided written informed consent before enrollment.

*(ii) Patient population:* The patient population consisted of 100 Caucasian patients undergoing bariatric surgery (BAS) at the 2nd Dept. of Surgery, Alfried-Krupp Hospital, Essen, Germany between 2010 and 2017. All patients met the following criteria for surgical weight loss therapy established by the NIH consensus conference [18], i.e., age >18 yrs., BMI ≥40 or ≥35 kg/m^2^, with co-morbidities, failure of medical weight loss, absence of medical or psychological contraindications for BAS, and evaluation by a multi-disciplinary team of medical, nutrition, psychiatry, and surgical specialists. Further demographic and clinical data included gender, liver enzymes, and metabolic parameters. Patients aged ≤18 or >65 years with liver pathologies other than NAFLD, history of organ transplantation, history of malignancy within the previous five years, alcohol abuse (average daily consumption >20 g/day for women or >30 g/day for men), drug abuse within the previous year, autoimmunity or genetic disorders, and therapy with immunosuppressive or hepatotoxic agents were excluded. Further excluded were patients with other known causes of secondary fatty liver disease (e.g., viral hepatitis, metabolic liver disease, or toxic liver disease). For details, see Appendix A. BAS was carried out laparoscopically. Briefly, operations were either performed as Roux-en-Y gastric bypass, sleeve gastrectomy, or gastric banding according to the surgeons’ decision.

*(iii) Histological assessment:* Histological evaluation of NAFLD and classification as NAFL or NASH according to the NAFLD activity score (NAS; range: 0–8) (Appendix A) were performed by two experienced pathologists according to Kleiner et al. [19].

### 2.2. Primary Human Hepatocytes

*(i) Human liver tissue:* Human liver tissue sections were obtained from partial liver resections or hemihepatectomies routinely performed at the Department of General, Visceral, Vascular, and Transplantation surgery of Otto-von-Guericke University, Magdeburg, Germany. Resected liver tissue was transferred to the Institute of Pathology and Neuropathology Otto-von-Guericke University, Magdeburg, Germany, for assessment and only tumor- and metastasis-free liver tissue was utilized for isolation of primary human hepatocytes (PHH).

*(ii) Isolation of primary hepatocytes:* PHH were isolated as described before [20]. Briefly, liver tissue was dissolved by perfusion with collagenase type IV at 40 °C. Cell suspensions were filtered through a 4 μm mesh, and hepatocytes were sedimented by centrifugation at RT and 50× *g*. Cells were resuspended in DMEM/Ham’s F-12 with 10% heat-inactivated (30 min; 56 °C) fetal bovine serum (FBS), 100 U/mL penicillin, 0.1 mg/mL streptomycin, and 2 mM l-glutamine, seeded at 1 × 10^6^ cells/cm^2^ in collagen coated 6-well plates (Nunc, Roskilde, Denmark; ThermoFisher Scientific, Oberhausen, Germany) and incubated for 24 h at 37 °C, 5% CO_2_, 95% RH before treatment, as described below.

### 2.3. HepG2 Hepatoblastoma Cells

The HepG2 cell line (ATCC Cat# HB-8065, RRID:CVCL_0027) (CLS Cell Lines Service, Eppelheim, Germany) was cultured in DMEM high-glucose medium (Invitrogen, Carlsbad, CA, USA) with 10% heat-inactivated (30 min; 56 °C) FBS (formerly Sigma-Aldrich/now: Merck, Darmstadt, Germany), 100 U/mL penicillin, 0.1 mg/mL streptomycin, and 2 mM l-glutamine, seeded at 3.5 × 10^4^ cells/cm^2^ in cell culture flask, or 6-well, or CellStar 96-well plates (Merck, Darmstadt, Germany) according to the experimental set-up and incubated for 24 h at 37 °C, 5% CO_2_, 95% relative humidity. Subsequently, HepG2 cells were handled as detailed below under Section 2.5
*LOLA Treatment in Models of Steatosis, Insulin Resistance, and Metabolic Syndrome*. (All other reagents from, formerly, Gibco/now: ThermoFisher Scientific, Oberhausen, Germany).

### 2.4. Viability

Appropriately diluted suspensions of primary hepatocytes or HepG2 cells were admixed with equal volumes of 0.4% trypan blue solution and incubated for 1 min at RT. Immediately after incubation, unstained viable cells were counted by light microscopy using a hemocytometer. Cells were then seeded at viabilities of 95–99% at the densities specified for PHH or HepG2 cells, respectively.

### 2.5. LOLA Treatment in Models of Steatosis, Insulin Resistance, and Metabolic Syndrome

Initially, PHH or HepG2 cells were starved for 12 h. Afterwards cells were incubated for 24 h with (i) 1 mM 2:1 oleate:palmitate long-chain free fatty acids (FFA) for mimicking a steatosis-like state; (ii) 50 nM insulin for inducing insulin resistance; or (iii) a combination of both as an in vitro model to approximate MetS-like conditions. After washing the cells with PBS, they were subsequently treated for 24–48 h with LOLA (Merz Pharma, Frankfurt/Main, Germany) at different concentrations (0, 20, 40, 60, 80, 100, 200, 400 mM) in culture medium with 5% FCS. All experiments were carried out in triplicates. (All reagents from Merck, Darmstadt, Germany).

### 2.6. Proliferation, Cytotoxicity and Cell Death

Proliferation or cytotoxicity, respectively, were measured by two confirmatory colorimetric assays, i.e., the CCK-8 assay and the MTT assay (both: formerly Sigma-Aldrich, now Merck, Darmstadt, Germany). Cell death markers M65 (for overall cell death) and M30 (for apoptosis) were assessed in culture supernatants using the M65 and the M30 Apoptosense enzyme-linked immunosorbent assays (ELISAs) (Peviva VLV bio, Nacka, Sweden). For quantifying apoptotic cells based on activated caspase 3/7, we employed the fluorometric CellEvent^TM^ Caspase-3/7 Green Detection Reagent endpoint assay (Invitrogen). All assays were performed according to the manufacturers’ instructions.

*(i) CCK-8 assay:* Proliferation/cytotoxicity was determined with the highly sensitive colorimetric cell counting kit-8 (CCK-8). The assay was performed according to the manufacturer’s instructions. Briefly, 100 µL volumes of cells in culture medium were seeded at 10,000 cells/well in 96-well plates. Suspensions were admixed with 10 µL/well of CCK-8 solution and kept at 37 °C, 5% CO_2_, 95% relative humidity. During the 2 h incubation period, the reagent—WST-8 (2-(2-methoxy-4-nitrophenyl)-3-(4-nitrophenyl)-5-(2,4-disulfophenyl)-2H-tetrazolium, monosodium salt)—is reduced by cellular dehydrogenases to form a water-soluble orange formazan product whose amount is directly proportional to the number of viable cells. The formazan product was determined by the CLARIOstar^®^ *Plus* microplate reader (BMG Labtech, Ortenberg, Germany) at λ = 450 nm.

*(ii) MTT assay:* Proliferation/cytotoxicity was also determined with a colorimetric MTT assay by adhering to the manufacturer’s instructions. Briefly, 100 µL volumes of cells in culture medium were seeded at 10,000 cells/well in 96-well plates. Suspensions were admixed with 25 µL/well of MTT buffer [i.e., 3.5 mg/mL MTT (formerly Sigma-Aldrich/now Merck, Darmstadt, Germany) in *Aq. dest.*], and incubated for 60 min at 37 °C, 5% CO_2_, 95% relative humidity. Cellular NAD(P)H-dependent oxidoreductases reduce the tetrazolium dye, 3-(4,5-dimethylthiazol-2-yl)-2,5-diphenyltetrazolium bromide (MTT), to a water-insoluble purple formazan product. After incubation, 125 µL of MTT extraction buffer (i.e., 95% isopropanol, plus 5% acetic acid) was added, followed by another 20 min incubation at RT. The formazan product was then determined by the CLARIOstar^®^ *Plus* microplate reader at λ = 550 nm.

*(iii) Necrosis* vs. *apoptosis:* Upon cell death, cellular proteins, including cytokeratin 18 (CK18), are released. The M65 ELISA detects both cleaved and uncleaved CK18, thus covering the total amount of necrotic and apoptotic cells, whereas the M30 ELISA detects the CK18 K18Asp396 neoepitope, which is only exposed after apoptotic cleavage by activated caspase-3. Specifically, 20 µL samples, each, of cell culture supernatants were entered into the assays. In case of the M65 ELISA, a solid phase-bound monoclonal antibody captured CK18, whereupon the horseradish peroxidase (HRP)-conjugated monoclonal M65 antibody recognized another CK18 epitope. In the M30 Apoptosense ELISA, a solid phase-bound antibody also captured CK18, while the HRP-conjugated monoclonal M30 antibody specifically recognized the K18Asp396 neo-epitope (also known as K18Asp396-NE). In both assays, after 2 h at RT in an orbital shaker at 600 rpm, the unbound conjugate was then washed off, 200 µL TMB substrate was added for 20 min in the dark at RT, and the color development was stopped by adding the stop solution. The absorbance was measured using the CLARIOstar^®^ *Plus* microplate reader at λ = 450 nm. As the resulting color is directly proportional to the concentration of the analyte, its concentration (given in U/L) could be calculated from a standard curve measured in parallel.

*(iv) Caspase 3/7 Detection:* Briefly, cells (1 × 10^4^ cells/well in 96-well plates) were cultured, treated as indicated, media were removed, followed by adding 100 µL/well of detection reagent (5 µM) in PBS with 5% FCS to the cells, and incubated for 30 min at 37 °C. Fluorescence intensities [λ_EX_ = 502 nm; λ_EM_ = 530 nm] were then measured on a CLARIOstar^®^ *Plus* microplate reader. Apoptotic cells are green, while healthy cells remain unstained. The signal can be visualized under a microscope and/or measured, respectively.

### 2.7. Metabolism

*(i) Ammonia (NH_3_):* Amino acid metabolism produces NH_3_, which is converted into urea in the urea cycle that is primarily located in the liver. After collection of the cell culture supernatants, cells were spun to remove debris and detached cells, and NH_3_ was quantitated in the supernatants. The fluorometric ammonia assay (Sigma-Aldrich) is based on the *o*-phthalaldehyde method in which the reagent reacts with NH_3_ or its cation, NH_4_^+^, respectively. Briefly, 10 µL/well of each standard was pipetted into a 96-well plate. Likewise, 10 µL/well of each test sample was plated into separate wells. Ninety µL/well of the freshly prepared working reagent mix was added to each standard or sample. Plates were tapped immediately for mixing, and the reaction was incubated for 15 min in the dark at RT. The resulting fluorometric signal was detected with the CLARIOstar^®^ *Plus* microplate reader (λ_EX_ = 360 nm/λ_EM_ = 450 nm). The signal is proportional to the sample concentration of NH_3_ or NH_4_^+^, respectively.

*(ii) Adenosine-5′-Triphosphate (ATP):* ATP levels were determined in cell culture (2.5 × 10^4^ cells/well in white 96-well plates in 100 µL medium/well) using the Luminescent ATP Detection Assay Kit (Abcam, Cambridge, UK) according to the manufacturer’s instructions. After treatment as indicated, 50 µL detergent solution was added, and plates were shaken for 5 min at RT in an orbital shaker at 600 rpm for cell lysis in the culture wells. ATP was stabilized by irreversibly inactivating ATPases, thus ensuring that the signal obtained corresponds to the endogenous ATP levels. This step was followed by adding 50 µL substrate solution, shaking for 5 min in an orbital shaker at 600 rpm, and storing of the test plate for 10 min in the dark at RT. Luminescence was analyzed against ATP control standards (0.1 nM–10 µM) using the CLARIOstar^®^ *Plus* microplate reader.

*(iii) Coenzyme A (CoA):* Determination of CoA in cell culture was performed by a fluorometric CoA assay (abcam) according to the manufacturer’s instructions. Samples of HepG2 total protein lysates (25 µL, each) were dissolved in 25 µL volumes of RIPA buffer, admixed with the CoA assay reaction mixture, and incubated for 30 min at RT. Fifty µL, each, of samples and standards were added per well, followed by 50 µL CoA assay reaction mixture to obtain total volumes of 100 µL/well, and measured with a CLARIOstar^®^ *Plus* microplate reader (λ_EX_ = 490 nm; λ_EM_ = 520 nm). Serial dilutions of the CoA standard (0.01–10 µM) were employed to determine the concentrations. The CoA standard curve was generated by plotting relative fluorescence units (RFUs) and serial dilutions of the CoA standard. CoA levels in the test samples were calculated from the calibration curve.

*(iv) Glutathione (GSH):* For quantifying cellular GSH levels, we employed the luminometric GSH-Glo™ Glutathione Assay (Promega, Madison, WI, USA) according to the manufacturer’s instructions. This assay is based on the conversion of the derivative, luciferin-NT, into luciferin in the presence of GSH, as catalyzed by glutathione *S*-transferase. Briefly, 1 × 10^4^ cells/well in 96-well plates (CellStar) were cultured, treated as indicated, media were removed, and the GSH-Glo™ reaction reagent (Luciferin-NT and Glutathione S-Transferase in GSH-Glo™ reaction buffer) was added. After 30 min incubation at RT, reconstituted Luciferin Detection Reagent was added, followed by 15 min incubation at RT, and GSH changes were determined in the same 96-well plates. Luminescence was measured using the CLARIOstar^®^ *Plus* microplate reader. The signal generated is proportional to the GSH amount in the sample.

### 2.8. Mitochondrial State

*(i) Mitochondrial membrane potential (ΔΨm):* To determine ΔΨm after cell culture, we performed a fluorometric JC-10 mitochondrial membrane potential assay kit (abcam) according to the manufacturer’s protocol. HepG2 cells were cultured on 96-well plates, and 30 min before the end of the respective treatments, 50 µL of JC-10 dye loading solutions were added to each well. After 30 min incubation, fluorescence intensities [λ_EX_ = 485 nm; λ_EM_ = 515 nm (cut-off at 515 nm)] and [λ_EX_ = 540 nm; λ_EM_ = 590 nm (cut-off at 570 nm)] were measured on a CLARIOstar^®^ *Plus* microplate reader. Changes of the ΔΨm were detected as the ratio between the aggregate (λ_EM_ = 515 nm) and monomeric forms (λ_EM_ = 590 nm) of JC-10. Increasing ratios indicate mitochondrial membrane depolarization. In some cases, ΔΨm data are displayed together with representative photomicrographs (*cf*. below, *Fluorescence Microscopy*).

*(ii) Mitochondrial morphology and superoxide production:* HepG2 cells were seeded into collagen-coated glass chambers. After treatments, cells were stained with 500 nM MitoTracker Green FM (ThermoFisher Scientific) for 30 min at 37 °C to visualize mitochondrial morphology. After three washing steps with cell culture media at RT, nuclei were counterstained with 8 μM Hoechst 33,342 (Invitrogen) for 30 min at 37 °C. Intramitochondrial superoxide radical generation was detected by MitoSOX Red (ThermoFisher Scientific). Once bound to nucleic acids, this fluorochrome is rapidly oxidized by superoxide to form a strongly red fluorescent dye (λ_EX_ = 510 nm; λ_EM_ = 580 nm), but not by any other reactive oxygen or nitrogen species. To this end, cells were incubated with 2.5 μM MitoSOX Red in 1 mL Hank’s balanced salt solution (HBSS) at pH 7.4 (ThermoFisher Scientific) for 10 min at 37 °C. Subsequently, cells were washed three times with HBSS at RT and surveyed by fluorescence microscopy (λ_EX_ = 350 nm; λ_EM_ = 461 nm).

### 2.9. Fluorescence Microscopy

Image acquisition for mitochondria was performed with a Zeiss AxioVert 200 M inverted microscope equipped with an Apochromat ×63/1.4, and Zeiss filter set No. 62, Colibri 2 (λ_EX_ = 370/40 nm, and λ_EM_ = 474/28 nm), as well as an AxioCam MRm monochrome digital camera for optical sectioning, and an ApoTome.2 to eliminate residual stray light. Optical data were analyzed by AxioVision imaging software package v4.8. Z-stacks of microscope images (i.e., 2D data) were acquired and turned into single 3D images to obtain an overall representation of the mitochondrial morphology. (All equipment from Carl Zeiss Microscopy, Jena, Germany.)

### 2.10. Total Protein

Primary hepatocytes or HepG2 cells were subjected to a standard procedure for obtaining extracts of total protein. To obtain total protein extracts, cells were lysed for 30 min at 4 °C in RIPA lysis buffer. Protein lysates were prepared using RIPA lysis buffer (25 mM Tris HCl pH 7.6, 150 mM NaCl, 1% NP-40, 1% sodium deoxycholate, 0.1% SDS containing complete EDTA-free protease and phosphatase inhibitor tablets, enabling broad-spectrum protection from proteolysis by serine proteases, cysteine proteases, aspartic acid proteases, aminopeptidases, and phosphatases). Lysates were centrifuged at 13,000× *g* for 30 min at 4 °C, and the resulting supernatants were stored at −80 °C for analysis. Protein concentrations were quantified colorimetrically by a bicinchoninic acid (BCA) assay according to the manufacturer’s instructions. This assay highly sensitively detects protein-dependent reduction of Cu^2+^ to Cu^+^ by BCA. All reagents and the BCA assay (from ThermoFisher Pierce Scientific, Oberhausen, Germany) contained complete mini EDTA-free protease and phosphatase inhibitor tablets. Lysates were centrifuged at 13,000× *g* for 30 min at 4 °C, and the resulting supernatants were stored at −80 °C for further analysis. Protein concentrations were determined by the BCA assay (all reagents and the BCA assay from ThermoFisher Pierce Scientific).

### 2.11. Immunoblotting

Either (i) 5 µg samples of total protein from PHH or (ii) 10 µg samples of total protein from HepG2 cells were dissolved in x1 Laemmli sample buffer for protein electrophoresis (Bio-Rad Laboratories; Feldkirchen, Germany) with 2-mercaptoethanol (Sigma-Aldrich), separated by SDS-PAGE (ThermoFisher Scientific), and transferred to nitrocellulose membranes (GE Healthcare, Munich, Germany). Immunoblotting was performed by standard procedures using primary antibodies and, subsequently, the appropriate horseradish peroxidase (HRP)-conjugated secondary antibodies. Antibodies are detailed in Appendix A. Bound antibodies were visualized using the SuperSignal^®^ West Dura Enhanced Chemiluminescence Substrate by Pierce (Rockford, IL, USA). Glyceraldehyde-3-phosphate dehydrogenase (GAPDH) or β-actin, respectively, were employed as loading controls.

### 2.12. Immunofluorescence

HepG2 cells were seeded at 3.5 × 10^4^ cells/cm^2^ in 4-well Nunc™ Lab-Tek™ II Chamber Slide™ Systems (Nunc) and cultured in DMEM high-glucose medium (Invitrogen) with 10% heat-inactivated FBS (Sigma-Aldrich), 100 U/mL penicillin, 0.1 mg/mL streptomycin, and 2 mM l-glutamine, and incubated for 24 h at 37 °C, 5% CO_2_, 95% relative humidity (all reagents from Gibco). After treatments, cells were washed twice with phosphate-buffered saline (PBS) at RT and fixed with pre-cooled (−20 °C) methanol for 10 min at 4 °C. Cells were washed ×3 with PBS at RT, and permeabilized for 5 min with 0.1% Triton X-100 in PBS. Nonspecific binding sites were blocked with 10% FBS in PBS for 30 min at RT. Cells were then incubated for 2 h at RT with the respective primary rabbit anti-human antibodies at 1:100 dilution. After three washes with PBS at RT, samples were incubated in the dark with Alexa Fluor 488-conjugated anti-rabbit antibody for 1 h at 1:100 dilution, rinsed (×3: PBS; ×1: H_2_O), and mounted in Prolong Gold Antifade mounting medium with 4′,6-diamidino-2-phenylindole (DAPI) (all reagents from ThermoFisher Scientific). Cells were evaluated by fluorescence microscopy. For antibodies, see Appendix A.

### 2.13. Quantitative Real-Time Polymerase Chain Reaction (qRT-PCR)

Total RNA from cultured cells was isolated with the RNeasy Mini kit. Isolated and purified RNA was transcribed into cDNA (for primers, see Appendix A) with the QuantiTect RT Kit using 1 µg of total RNA, each, following the manufacturers’ instructions (all materials from Qiagen, Hilden, Germany). Qrt-PCR was performed using a CFX96 Touch qRT-PCR Detection System (Bio-Rad Laboratories, Munich, Germany) using the QuantiTect SYBR Green Kit (Qiagen, Hilden, Germany) in volumes, each, of 15 μL including 2 μL of the appropriate cDNAs. Primers are detailed in Appendix A. Melting curves were collected to ascertain the specificity of the PCR-products. Changes in mRNA-expression were calculated by the ΔΔ-Ct method and are presented as fold change in relation to the expression of a reference gene, i.e., either hypoxanthine phosphoribosyltransferase (HPRT) or succinate dehydrogenase complex, subunit A (SDHA). Relative gene expressions were calculated from the threshold cycles in relation to the respective reference gene and to the experimental control samples.

### 2.14. Statistics

If not stated otherwise, data represent ≥3 independent experiments. Statistical significance was determined either by unpaired (or paired, when applicable) 2-tailed *t*-test, Mann–Whitney U test for non-parametrical data, or in experiments with >2 conditions by 1-way ANOVA, with Tukey’s post hoc test for individual experimental conditions. For experiments with two independent variables (i.e., metabolic conditions and LOLA treatment), 2-way ANOVA with Tukey’s post hoc test was applied. All analyses were performed with Prism 7 (GraphPad Software, San Diego, CA, USA). Significance was assumed at *p* ≤ 0.05. If not stated otherwise, data are presented as means ± SD or ±SEM, respectively.

## 3. Results

### 3.1. Patient Cohort

To assess whether potential targets of LOLA are altered in NASH and thus could serve as therapeutic points of action, mRNA expressions of potential target genes were assessed in liver tissue of morbidly obese patients. All NAFLD patients were grouped according to histological evaluation as NAFL (steatosis, NAS ≤ 4) or NASH (NAS ≥ 5). For additional information, we refer again to Appendix A.

### 3.2. Cellular Models

For mechanistic analysis of effects caused by LOLA treatment, two in vitro models were employed. First, primary human hepatocytes (PHH) isolated from native human liver tissue were treated with different concentrations of LOLA with subsequent analysis of mRNA expression of various target genes associated with fatty acid transport and synthesis as well as with BCAA catabolism.

Additional experiments were performed in the hepatoblastoma cell line HepG2 (e.g., Figure 1) by treating these cells with FFAs (1 mM of 2:1 oleate:palmitate long-chain FFAs) as a model for steatosis; insulin (50 nM) to reflect insulin resistance; or FFAs combined with insulin to mimic conditions of the metabolic syndrome. In each of these conditions, HepG2 cells were treated with different LOLA concentrations.

### 3.3. In Vitro Treatment with LOLA Appears Safe at Up to 40 nM and Results in ATP Decrease

To determine a suitable dose of LOLA for treating HepG2 cells, their proliferation was determined. We found a stable proliferation rate in the range of 0 mM to 40 mM LOLA. A drop in the proliferation rate of ~30% was detected at 60 mM and was largely maintained at higher doses of LOLA, which may indicate cytotoxicity. Notably, inter-sample variances at all of the lower LOLA concentrations were considerably smaller than those observed with the higher concentrations (Figure 1). In parallel, LOLA concentration-dependently effected a decrease in hepatocellular ATP production (Figure 1).

### 3.4. LOLA Enhances Protein Expression of Urea Cycle Enzymes and Reduces NH_3_/NH_4_^+^ Release in HepG2 Cells under Conditions of Metabolic Alteration

We found increased protein expression rates of urea cycle enzymes in hepatocytes enriched from liver tissue of patients with either NAFL or NASH (Figure 2). Specifically, immunoblots showed that the expression rates of both (Figure 2A) carbamoyl phosphate synthetase (CPS1) and (Figure 2B) ornithine transcarbamylase (OTC)—both normalized against glyceraldehyde-3-phosphate dehydrogenase (GAPDH)—were significantly elevated in hepatocytes from NASH patients when compared with those obtained from NAFL patients. Thus, it is evident (i) that NAFLD is associated with the upregulation of CPS1 and OTC expression when compared with non-NAFLD hepatocytes and (ii) that the expression of urea cycle enzymes increases further upon inflammatory exacerbation of the disease.

Ammonia produced by untreated HepG2 cells and those under in vitro conditions mimicking steatosis, insulin resistance, or the MetS did not differ significantly. However, treatment with LOLA consistently reduced the amount of NH_3_/NH_4_^+^ release into the culture supernatants in a concentration-dependent manner under all these conditions (Figure 2C).

### 3.5. Reduced Enzyme Expression for BCAA Catabolism in NASH Is Counteracted by LOLA

BCAAs are increased in the sera of NAFLD patients [21,22,23]. We therefore hypothesized that LOLA may act beneficially on BCAA catabolism. Here, we show that hepatocytes from NASH patients exhibited reduced mRNA expressions of enzymes of BCAA catabolism when compared to NAFL (Figure 3A,B). In PHH, we then investigated the expression rates of the mRNAs for all three enzymes of BCAA catabolism, i.e., branched-chain amino acid aminotransferase 2 (*bcat2*), branched-chain keto acid dehydrogenase (*bckdha*), and branched-chain keto acid dehydrogenase kinase (*bckdk*): we could demonstrate that LOLA dose-dependently increased the expression rates of these transcripts (Figure 3(C1–C3)), but this effect remained statistically insignificant. Whether LOLA might contribute to the reconstitution of BCAA metabolism in patients with NAFL or NASH thus will have to be investigated more closely.

### 3.6. LOLA Modulates Gene Expressions Related to Fatty Acid Import, Synthesis, and Regulation

Since effects of LOLA on fatty acid transport and synthesis as well as mitochondrial biogenesis have not yet been investigated, we measured mRNA and protein expressions of transport proteins, enzymes and a transcription coactivator involved in these processes (Figure 4). The fatty acid translocase CD36 (now classified as scavenger receptor B2) [24] is an integral membrane protein responsible for the cellular import of (long-chain) FFAs. LOLA treatment strongly downregulated *cd36* mRNA expression in the HepG2 models of steatosis and MetS (Figure 4A). Fatty acid synthase (FASN) catalyzes the formation of palmitate from acetyl-CoA and malonyl-CoA. LOLA treatment in PHH downregulated the expression of *fasn* mRNA (Figure 4B). Interestingly however, FASN protein expression remained unaffected by LOLA under all conditions tested (Figure 4C). Conversely, LOLA in PHH induced the expression of *scd1* mRNA (Figure 4D), which encodes for stearoyl-CoA desaturase-1 (SCD1) as a key enzyme in the generation of monounsaturated fatty acids. Acetyl-CoA carboxylase 1 (ACC1) catalyzes the carboxylation of acetyl-CoA to malonyl-CoA, the rate-limiting step in fatty acid synthesis. In HepG2 cells under in vitro conditions mimicking steatosis, insulin resistance, and MetS, we found increased protein expression of ACC1, which was reduced by LOLA in the conditions of steatosis and insulin resistance (Figure 4C). In the liver, *srebf1* mRNA is translated into the hepatic protein isoform of sterol regulatory element-binding protein-1c (SREBP-1c) [25]. We observed elevated *srebf1* mRNA expression in HepG2 cells under MetS conditions. LOLA treatment reduced *srebf1* mRNA expression under conditions of steatosis and—less consistently—MetS (Figure 4E). *Cpt1*/carnitine palmitoyltransferase 1 (CPT1) shuttles long-chain fatty acids across the mitochondrial membrane for subsequent β-oxidation. In PHH, LOLA treatment had differential concentration-dependent effects on the expression of *cpt1* mRNA (Figure 4F). Peroxisome proliferator-activated receptor-γ coactivator (PGC)-1α, a transcription coactivator that plays a key role in regulating cellular energy metabolism, stimulates mitochondrial biogenesis. LOLA did not have a favorable effect on the transcription of *pgc1α* (Figure 4G).

### 3.7. LOLA Reduces Cellular Production of Acetyl-Coenzyme A and ATP in HepG2 Cells

Under all conditions—i.e., in untreated HepG2 cells and in the in vitro models of steatosis, insulin resistance, and MetS—we found a dose-dependent reduction of cellular ATP production in the presence of LOLA (Figure 5A). Notably, we observed a stronger decrease in ATP than had been detected in the experiment depicted in Figure 1. Nevertheless, the LOLA-dependent decrease in ATP production is in line with the similarly LOLA-dependent reduction in acetyl-coenzyme A (acetyl-CoA) (Figure 5B).

### 3.8. LOLA Induces Phosphorylated AMP-Activated Protein Kinase α upon Metabolic Alteration

With the exception of the untreated control, in the models of steatosis, insulin resistance, and MetS, LOLA effected consistent and dose-dependent increases in phosphorylated AMP-activated protein kinase α (p-AMPK-α) (Figure 5C), which acts as a coordinator of signals controlling cellular energy balance.

### 3.9. LOLA Reconstitutes the Mitochondrial Membrane Potential while Avoiding Superoxide Induction

In order to assess whether the observed effects of LOLA in HepG2 cells might be related to altered energy regulation and mitochondrial performance and integrity, the mitochondrial membrane potential (ΔΨm) and superoxide production were measured in vitro (Figure 6). In HepG2 cells, ΔΨm was found to be significantly depolarized by the in vitro conditions of steatosis and MetS when compared to untreated controls (Figure 6A). In contrast, insulin treatment alone had no clear effect. In FFA-treated and MetS-conditioned HepG2 cells, LOLA concentration-dependently reconstituted ΔΨm (Figure 6B). Importantly, the mitochondrial membrane integrity appeared unaffected by LOLA treatment (Figure 6C).

Regardless of the applied concentration, LOLA did not induce mitochondrial superoxide production in HepG2 cells. However, we observed slight changes in the intracellular locations and shapes of the mitochondria (Figure 7).

## 4. Discussion

This study aimed to identify novel molecular targets of l-ornithine-l-aspartate as a potential treatment option for NAFLD. We demonstrated that LOLA concentration-dependently reduced the cellular release of NH_3_; normalized the expression of catabolic BCAAs enzymes; beneficially modulated targets of fatty acid import/transport, synthesis, and regulation; decreased cellular acetyl-CoA and ATP; favorably modulated the expression of master regulators and genes of energy balance/mitochondrial biogenesis; and reconstituted the mitochondrial membrane potential without overt side-effects. We believe that these results warrant further investigations to establish LOLA as a treatment option for the ever-increasing medical challenge of NAFLD worldwide. Thomsen and colleagues recently highlighted the important role of NH_3_ in NAFLD: besides its deleterious effects on other organs, hepatic NH_3_ accumulation leads to inflammation, stellate cell activation and fibrogenesis, which may play a key role in the facultative NAFL → NASH → cirrhosis → hepatocellular carcinoma exacerbation cascade. The authors therefore concluded that NH_3_-lowering treatment strategies (which we believe include LOLA) should be explored clinically in patients with NAFLD [26].

The main targets for the effect of LOLA according to current evidence are the urea cycle enzymes CPS1 and OTC, whose expression was elevated in hepatocytes from NASH patients when compared to those from NAFL patients. In contrast, De Chiara et al. [27] found the hepatic expression of OTC to be reduced in NASH when compared to NAFL. Moreover, the hepatocellular promoter regions of the *otc* gene in NALF and NASH patients, and the *csp1* and *otc* promoters in adipose mice were hypermethylated [27]. We thus cautiously hypothesize that our unexpected finding may relate to the fact that while we measured the expression of these enzymes, the proteins themselves may be functionally impaired. If this assumption can be substantiated, increased enzyme expression rates in NAFL—and even more so in NASH—may indicate a cellular response to compensate for such functional impairment.

We recently evidenced that the catabolism of BCAAs is impaired and/or downregulated in NAFLD [28]. Our in vitro results now suggest that LOLA may reconstitute the breakdown of BCAAs in patients with NAFLD via increasing the expression of enzymes of BCAA catabolism. In primary hepatocytes, LOLA concentration-dependently triggered elevated transcription rates of all BCAA-relevant enzymes. In parallel to reducing increased serum BCAA concentrations in NASH patients, enhanced BCAA catabolism in hepatocytes would theoretically increase the concentrations of the products glutamate and acetyl-CoA for further metabolization via the urea cycle. This would contribute to improved detoxification of NH_3_. However, NASH patients revealed higher mRNA expression and slightly lower protein expression of BCAA-catabolizing enzymes. Thus, the mechanisms by which LOLA directly affects enzyme activity will have to be analyzed in future studies.

Among the possible additional target genes of LOLA analyzed in this study were FA transporters, FA-synthetizing enzymes, and regulators of lipid metabolism. LOLA treatment resulted in strong downregulation of *cd36* mRNA expression in HepG2 cells under metabolic alteration. In a previous work, we could show that CD36 upregulation in morbidly obese patients with NASH is associated with apoptosis [29]. Thus, LOLA treatment might exert a liver-protective role in such patients. CD36 may also be involved in diabetes, glucose intolerance, and atherosclerosis [30], so that LOLA may have additional beneficial effects on different manifestations of metabolic dysregulation including the MetS. The multi-enzyme protein FAS is the master regulator of FA synthesis. Significant downregulation of the *fasn* mRNA transcript by LOLA supposedly diminishes the degree of de novo lipogenesis, which would slow down progression of NAFLD. In contrast, LOLA concentration-dependently increased the transcription of *scd1* mRNA, which encodes for stearoyl-CoA desaturase 1. SCD1 catalyzes the rate-limiting step in the formation of monounsaturated fatty acids. We believe that this finding will require further investigation as constitutively (over)active SCD1 is a biochemical hallmark of cancer cells [31]. Also, SCD1 overexpression and monounsaturated fatty acids, as a product of active SCD1, are involved in the development of hypertriglyceridemia, diabetes, and atherosclerosis [32]. It thus needs to be clarified whether LOLA-treatment of NAFLD patients permanently upregulates SCD1 expression and alters the concentrations of monounsaturated fatty acids. Sterol regulatory element-binding protein-1 plays a key role in the induction of lipogenesis [33]. In the liver, *srebf1* mRNA is translated into the hepatic protein isoform SREBP-1c [25]. Activation of mTORC1 by insulin leads to increased SBREP-1c production, which facilitates FFA storage as triglycerides [34]. Clinically, high blood levels of insulin in insulin resistance activates SREBP-1 that leads to hepatic steatosis [35]. This process can be inhibited by the suppression of SREBP-1c via sirtuin 1 [36], which protects it from NAFL. LOLA clearly and concentration-dependently reduced *srebf1* mRNA expression in the in vitro models of steatosis and insulin resistance. This can therefore be expected to reduce available active SREBP-1c protein and possibly diminish steatosis induction and insulin resistance [35]. This notion is supported by findings in mice in which feeding of a high-carbohydrate diet induced the expression of the hepatic *scd1* gene via an insulin-mediated SREBP-1c-dependent mechanism, which upregulated the synthesis of monounsaturated fatty acids and triglycerides in the liver [37]. Overall, LOLA appears to alter mRNA expressions of genes related to FA transport, synthesis, and regulation in a manner potentially beneficial to NAFLD patients.

Since the fundamental causes of NAFLD are excess calories and pathophysiological consequences of storage and metabolism of excess nutrients and calories, the regulation of energy expenditure may be an obvious and logical target for treating NAFLD. Acetyl-CoA is a key coenzyme in, e.g., the degradation of lipids (followed downstream by fatty acid β-oxidation), carbohydrates, and proteins, which eventually produces ATP. Therefore, therapeutic modulation of acetyl-CoA carboxylase is on the developmental agenda as a putative future treatment for NAFLD [38]. We now found that LOLA downregulated both acetyl-CoA and the production of ATP in HepG2 cells under conditions of steatosis, insulin resistance, and MetS. Increased turnover of ATP and acetyl-CoA in functionally intact mitochondria indicates increased energy consumption, which is considered beneficial for addressing NAFLD. Second, as already stated above—and here extended to the in vitro models of steatosis—insulin resistance, and MetS, the LOLA-induced cellular low-energy status might be beneficial for controlling HCC. In parallel, LOLA treatment also resulted in a consistent and dose-dependent increase in AMP-activated protein kinase α in all in vitro models of metabolic dysregulation. By means of the AMPK-α/p-AMPK-α ratio, this enzyme acts as a molecular sensor for hunger vs. satiation, hence controlling the cellular energy balance. Specifically, in response to binding AMP and ADP indicating a low cellular energy status, AMPK-α is activated by phosphorylation. It then stimulates hepatic fatty acid β-oxidation and ketogenesis and, conversely, inhibits cholesterol synthesis, lipogenesis, fatty acid and triglyceride synthesis, as well as adipocyte lipogenesis and lipolysis [39,40]. The consistent increase in p-AMPK-α under LOLA treatment in all conditions of metabolic injury suggests that LOLA may achieve an effect similar to the fasting state and may counteract insulin resistance; both of which would be highly desirable in the case of NAFLD. The mitochondrial membrane potential is generated by proton pumps and represents an essential component of energy storage during oxidative phosphorylation. Together with the H^+^ gradient, ΔΨm forms the transmembrane potential of H^+^ ions, which is harnessed to generate ATP. Long-lasting de- or hyperpolarization of ΔΨm may therefore lead to cell death and are causes of various pathologies [41]. In our experimental models of steatosis, insulin resistance, and MetS, LOLA was able to reconstitute ΔΨm, which supports the applicability of LOLA not only in NAFLD but also other chronic liver diseases with impaired mitochondrial integrity. As a cautionary note, statistical significance was only reached at 60 mM of LOLA, which may be cytotoxic (Figure 1). However, our results clearly demonstrated a concentration-dependent trend for LOLA towards reconstituting the mitochondrial membrane potential; future studies in sufficient sample sizes of patient-derived primary hepatocytes (as opposed to the HepG2 cell line) will therefore have to substantiate this encouraging finding and establish a suitable concentration for LOLA to be applied under real-world conditions in patients with NAFLD. Specifically, ΔΨm and the proton gradient play key roles in the maintenance of mitochondrial functionality and homeostasis [41]. Therefore, LOLA-mediated reconstitution of ΔΨm in NAFLD might provide greater ‘biochemical buffer capacity’ to an impaired organ in which many important metabolic processes directly or indirectly depend on mitochondrial integrity. Regardless of its concentration, LOLA did not induce mitochondrial superoxide production in the HepG2 hepatoblastoma cell line. However, we observed changes in the locations and shapes of the mitochondria. Whether these changes are of functional relevance remains to be elucidated. Still, one hypothetical scenario is that the LOLA-dependent reconstitution of ΔΨm may cause a transient reorganization of the mitochondria via rearranging the affected cells’ microtubules and/or actin cytoskeleton as might be expected from findings in other settings [42,43,44]. Testing this hypothesis will require prolonged experimental observation periods. Taken together, our results imply that LOLA treatment enhances energy expenditure and might mimic signaling given in a fasting state, which would benefit patients with NAFLD by compensating a state of continuous excess calorie supply.

LOLA definitely is an agent with an obviously fascinating spectrum of effects. Against this background, the question arises as to which individual aspects of this spectrum (if not all of them) are LOLA-specific. Since a range of therapeutic targets of LOLA has been identified, a follow-up study needs to compare the effects of LOLA not only with those of its singular components l-ornithine and l-aspartate, but also with representatives of other amino acid classes. As for the latter, a selection of hydrophobic, neutral and hydrophilic amino acids should be employed. Furthermore, these should have basic vs. acidic properties, be charged or uncharged, and have a polar or non-polar structure, respectively. In this way, it should be possible to narrow down whether the observed properties of LOLA (i) are specific for this agent, (ii) can be (partially) induced by other amino acids as well, and (iii) might also identify discrete molecular properties that are responsible for the observed effects.

We here presented multiple novel findings related to hepatotropic effects of l-ornithine-l-aspartate initially based on experiments in human primary hepatocytes and consolidated by studying various effects of LOLA on the HepG2 cell line. The resulting encouraging body of evidence provides a promising foundation for further in-depth investigations. We believe that such studies will corroborate the benefits of LOLA towards achieving our dedicated goal of establishing this compound as an efficacious medication for preventing and/or treating NAFLD, which is in line with the call by Thomsen and colleagues to clinically develop such medications [26]. Given the ever-increasing incidence of NAFLD as a metabolically associated pandemic in the face of a paucity of treatment options, employing LOLA as a metabolic modulator may significantly contribute to a safer, more effective, gentle, and less expensive management of this disorder.

## Figures and Tables

**Figure 1 pharmaceutics-16-00506-f001:**
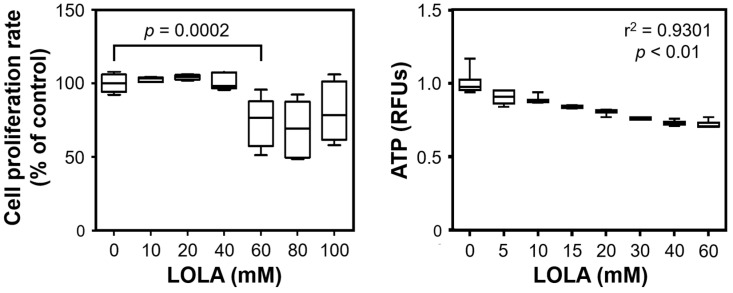
HepG2 proliferation and ATP production in the absence and presence of LOLA (all conditions: n = 3; arithmetic means ± SD; statistical significance by 2-way ANOVA with Tukey’s post hoc test).

**Figure 2 pharmaceutics-16-00506-f002:**
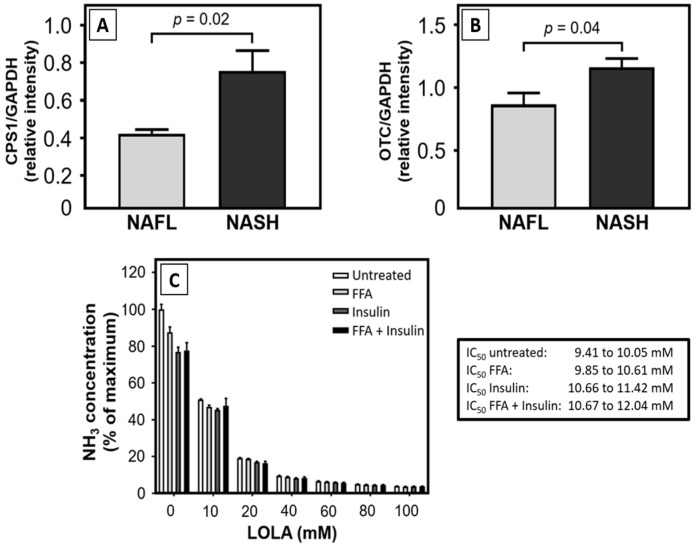
Expression of enzymes of the urea cycle NAFLD patients, and dose-dependent effects of LOLA on the production of NH_3_ in HepG2 cells in vitro. In NAFLD patients, (**A**) immunoblots showed that the expression rates of both (**A**) carbamoyl phosphate synthetase (CPS1) and (**B**) ornithine transcarbamylase (OTC)—both normalized against glyceraldehyde-3-phosphate dehydrogenase (GADPH)—were significantly higher in NASH than in NAFL (arithmetic means ± SD; statistical significance by Student’s *t* test). (**C**) Dose-dependent effects of LOLA on the production of NH_3_ in HepG2 hepatoblastoma cells and in in vitro models of steatosis (FFA), insulin resistance (insulin), and metabolic syndrome (FFA + insulin) compared to untreated controls (all conditions: n = 3; arithmetic means ± SD; statistical significance by 2-way ANOVA with Tukey’s post hoc test).

**Figure 3 pharmaceutics-16-00506-f003:**
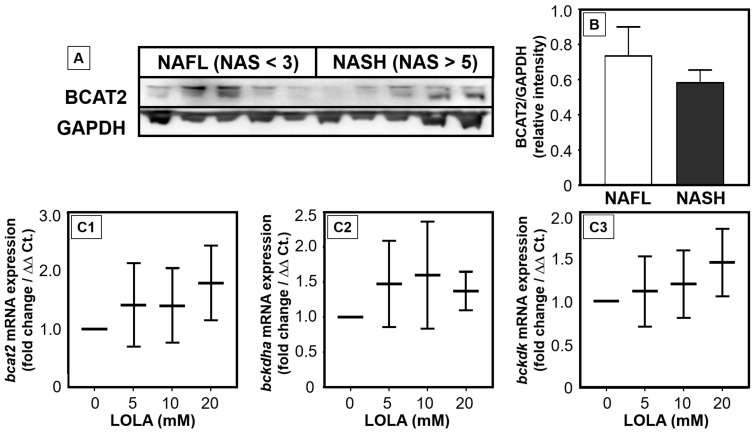
Expression of enzymes catabolizing branched-chain fatty acids (BCAAs) in patients with NAFLD and in primary human hepatocytes under LOLA treatment. (**A**) Representative immunoblot of BCAT2 and control GAPDH and (**B**) the complete results of GAPDH-normalized BCAT2 expression in patients with NAFL and NASH (n = 5, each). (**C**) In primary human hepatocytes, LOLA concentration-dependently upregulated the transcription of all key enzymes of BCAA catabolism, i.e., the mRNAs (**C1**) *bcat2*, (**C2**) *bckdha*, and (**C3**) *bckdk* (all conditions: n = 2; with means and ranges).

**Figure 4 pharmaceutics-16-00506-f004:**
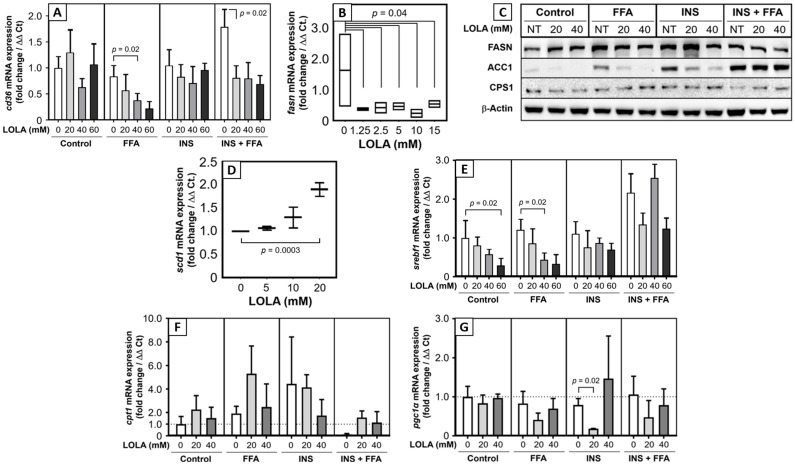
Expression of mRNAs (and one protein) relevant for cellular fatty acid import, de novo lipogenesis and mitochondrial biogenesis in primary human hepatocytes, and in in vitro models of steatosis (FFA), insulin resistance (INS), and metabolic syndrome (INS + FFA) of HepG2 cells. Shown are the mRNA expressions of (**A**) fatty acid translocase CD36; (**B**) fatty acid synthase (FASN) [as well as (**C**) FASN protein expression]; (**D**) stearoyl-CoA desaturase-1 (SCD1); (**E**) sterol regulatory element-binding transcription factor 1 (SREBF1) [in the liver translated into sterol regulatory element-binding protein-1c (SREBP-1c)]; (**F**) carnitine palmitoyltransferase 1 (CPT1); and (**G**) peroxisome proliferator-activated receptor-γ coactivator-1α (PGC-1α). For analysis, see text [all conditions: n = 3; arithmetic means ± SD; statistical significance by 1-way (PHH data) or 2-way (HepG2 data) ANOVA with Tukey’s post hoc test].

**Figure 5 pharmaceutics-16-00506-f005:**
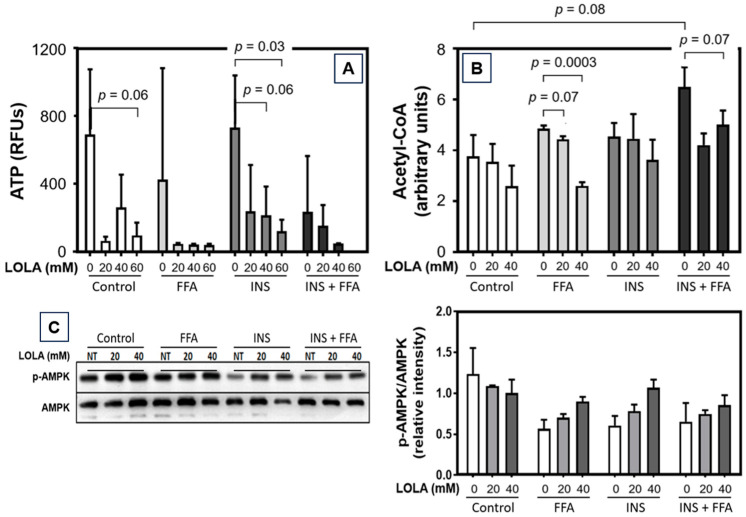
Dose-dependent effects of LOLA on the production of (**A**) ATP and (**B**) Acetyl-CoA in HepG2 cells under in vitro models of steatosis (FFA), insulin resistance (insulin), and metabolic syndrome (FFA + INS) compared to untreated controls (all conditions: n = 3; arithmetic means ± SD, and 2-way ANOVA with Tukey’s post hoc test was used for the statistical analyses of these experiments). (**C**) Dose-dependent effects of LOLA in untreated HepG2 cells and in in vitro models of steatosis (FFA), insulin resistance (INS), and metabolic syndrome (INS + FFA) on the expression of AMP-activated protein kinase α (AMPK-α), a key mediator of energy balance and mitochondrial biogenesis. An exemplary immunoblot is depicted on the left (all conditions: n = 3; arithmetic means ± SD).

**Figure 6 pharmaceutics-16-00506-f006:**
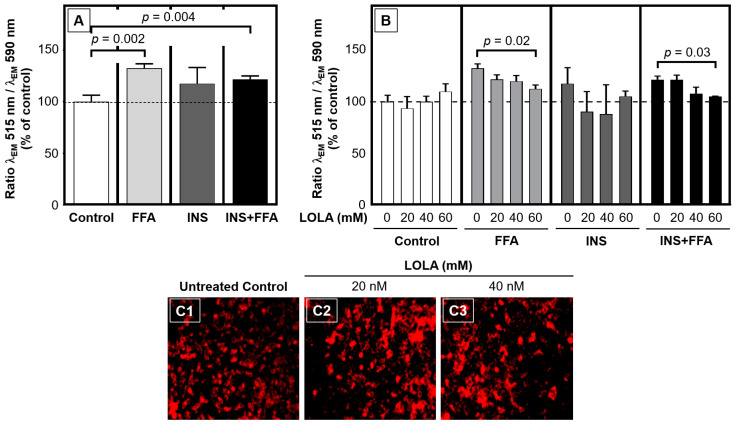
Mitochondrial membrane potential (ΔΨm) in HepG2-based in vitro conditions of steatosis (FFA), insulin resistance (INS), and metabolic syndrome (INS + FFA) are normalized by LOLA. (**A**) Compared to an untreated control condition set at 100%, ΔΨm was depolarized in the models of steatosis, insulin resistance, and metabolic syndrome. (**B**) LOLA concentration-dependently reconstituted ΔΨm. [(**A**,**B**) All conditions: n = 3; arithmetic means ± SEM; statistical significance by 2-way ANOVA with Tukey’s post hoc test.] (**C**) In fluoromicrographs of mitochondria stained with fluorescent dye JC-10, no leakage of the dye became apparent under LOLA treatment alone at different concentrations (**C2**,**C3**) when compared to the untreated control (**C1**).

**Figure 7 pharmaceutics-16-00506-f007:**
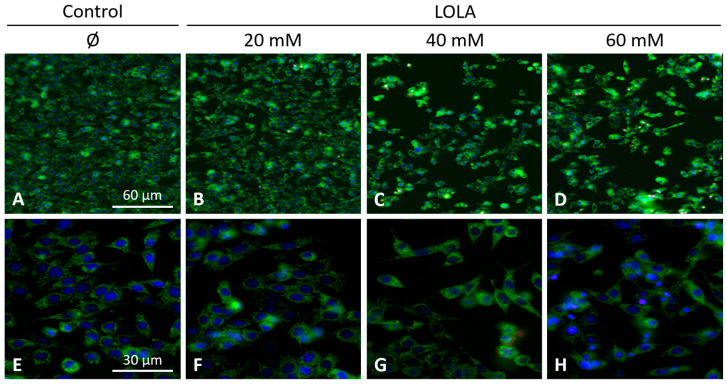
No superoxide production by HepG2 hepatoblastoma cells upon LOLA treatment. HepG2 cells were stained with MitoTracker Green FM (green) to visualize mitochondrial morphologies, and nuclei were counterstained with Hoechst 33,342 (blue). Intramitochondrial superoxide production was tracked by MitoSOX Red (red). When compared to the untreated control (**A**,**E**), no mitochondrial superoxide production was detected in the LOLA-treated conditions (**B**–**D**,**F**–**H**). Original magnifications: ×20 (upper row); ×40 (lower row).

## Data Availability

Data presented in this study are available on request from the corresponding authors. Certain data may not be available due to privacy restrictions.

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
