# Peer review of "l-Ornithine-l-Aspartate (LOLA) Normalizes Metabolic Parameters in Models of Steatosis, Insulin Resistance and Metabolic Syndrome"

_pharmaceutics, 2024, doi:10.3390/pharmaceutics16040506_

Round 1
Reviewer 1 Report
Comments and Suggestions for Authors
The topic of the manuscript, namely studying effects of the drug LOLA to correct metabolism in steatotic/insulin resistant hepatocytes is in principle interesting, but then the data fall short of being convincing.
· This manuscript proposes that the drug LOLA increases expression of enzymes involved in catabolism of branched amino-acids. However, when levels of bcat2, bckdha, and bckdk mRNAs in hepatocytes treated with LOLA are compared to that of the untreated control, there is no indication that any of the effect reaches significance. Also, in Fig.3A , it does not appear that hepatocytes derived from NASH patients expresses significantly lower levels of bcat2 as compared to that of NAFL patients. In respect of rigor, the figure does not say the number of patients included in the analysis.
· The manuscript proposes that, under conditions to reproduce in vitro effects by steatosis/insulin resistance, LOLA has beneficial effects to modulate expression of genes related to fatty acid import/transport across mitochondrial membrane, such as that of cd36 and cpt1, respectively, and fatty acid synthesis such as fasn, scd1, ACC1, and regulation of lipid metabolism (srbf1). However, in the column graph showing LOLA modulation of CD36 in hepG2 cells treated to model steatosis and insulin resistance there are no standard deviations and no statistical analysis. In Fig.4, there is also no statistical analysis of WB data of ACC and FASN abundance, nor of mRNA data on abundance of cpt1 and Scd1, and no apparent significative change in abundance pgc1 mRNA under all conditions tested except for one (decreased pgc1 in “INS” hepatocytes treated with 20 mM LOLA). These data are insufficient to reach the conclusion that LOLA has beneficial effects to correct these parameters. The same is true for modulation of AMPK.
· The manuscript proposes that LOLA restores to normal values mitochondrial membrane potential decreased by excess FFA and insulin in HepG2 cells. However, outcomes reach significance only when cells are treated with the highest concentration (60 mM) of LOLA, which appear to be cytotoxic in Fig. 1. In respect of feasibility of using LOLA he concentration of the drug used here is stellar (millimolar range).
Minor:
· Panel C is not indicated in Fig.2. Also, the small table is difficult to understand. What is the p value referring to?
· Please add letters to designate all panels of Fig.4 and revise figure legend)
Reviewer 2 Report
Comments and Suggestions for Authors
The authors have performed impressive experiments looking at multiple aspects of hepatocyte activity and metabolism to assess the possible effect of L-ornithine- L-aspartate on hepatocytes arising from patients with non-alcoholic fatty liver disease. The breadth of experiments is outstanding, including measurement of multiple metabolites and assessing changes in gene expression. Furthermore, the authors have assessed mitochondrial activity and mitochondrial membrane potential. Many changes are noted in response to increasing concentrations of L-ornithine- L-aspartate.
The work is quite convincing to the reader not experienced with prior studies of L-ornithine- L-aspartate on hepatocytes. There is only one source of concern which is the control experiments. The effects of L-ornithine Laspartate which breaks down to L ornithine and L-aspartate are clearly described. What is lacking is the demonstration that equivalent concentration of other metabolically active amino acids would not accomplish similar changes. So what this reviewer would like to see are comparison experiments with several different classes of amino acids in varying concentration, for example, lysine, arginine, and tyrosine. Admittedly, performing all these experiments is now a heavy request. Perhaps the authors can address this issue by mentioning this line of study as a future goal. If they are willing to do this, the issue of amino acid comparisons should be discussed at the outset in the introduction and then reiterated in the discussion.
SPECIFIC AIMS
Beneficial effects of LOLA appear not restricted to patients with chronic hepatitis C with 73
an increased risk of advanced fibrosis or cirrhosis [15], yet not in acute liver failure [16]
THIS SENTENCE IS HARD TO UNDERSTAND
Reviewer 3 Report
Comments and Suggestions for Authors
This study by Gieseler and coworkers is aimed at L-ornithine-L-aspartate (LOLA) normalizes metabolic parameters in models of steatosis, insulin resistance, and metabolic syndrome. The authors intended to identify novel molecular targets of L-ornithine-L-aspartate as a potential treatment option for NAFLD. Therefore, they demonstrated that LOLA concentration-dependently reduced the cellular release of NH3 and normalized the expression of catabolic BCAAs enzymes which beneficially modulated targets of fatty acid import/transport, synthesis, and regulation in the mitochondrial events. The overall investigation would contribute to improved detoxification of NH3. However, NASH patients revealed higher mRNA expression and slightly lower protein expression of BCAA-catabolizing enzymes. The novel finding of hepatotropic effects of LOLA was studied on the HepG2 cell line. The findings from this study provided the benefits of LOLA as an efficient medication for the prevention or treatment of NAFLD. Thus, evidence from this study showed the occurrence of NAFLD as a metabolically associated pandemic in the face of a paucity of treatment options, using LOLA as a metabolic modulator might significantly contribute to safer, more effective, gentle, and less expensive management of this kind of disorders. The provided data and cited references are fine. Therefore, it can be accepted in "Pharmaceutics" publication.

Reviewer 4 Report
Comments and Suggestions for Authors
The manuscript presented herein investigated the hepatotropic effects of L-ornithine L-aspartate in terms of cell proliferation, ATP production, NH3 production, acetyl-CoA production, protein and gene expression regulation involved in urea cycle, amino acid catabolism and fatty acid import, synthesis, and regulation, as well as mitochondrial membrane potential with models of steatosis, insulin resistance and metabolic syndrome. The experiments were well executed and the writing is clear. The beneficial effect demonstrated in this study warrants a further-in-depth investigations in L-ornithine L-aspartate as a candidate ameliorating NAFLD.
Minor revisions:
1. Line 299, should it be 'complete protease free EDTA"?
2. Line 534, "expression of these enzymes" should be more appropriate since we were not actually measuring a rate which involves changes in varied time points.
Round 2
Reviewer 1 Report
Comments and Suggestions for Authors
Unfortunately, the concerns that this reviewer had for the first review of this manuscript, are remaining. In the abstract the conclusion:
1)"In primary hepatocytes from NAFLD patients, urea cycle enzymes CSP1 and ornithine transcarbamylase (OTC) increase, while the catabolism of branched-chain amino acids (BCAAs) decreases with disease severity" does nor rest on solid, reproducible results as because "the results remained statistically insignificant". If the Authors feel that this is nevertheless a promising result as they claim, they should prove that by a rigorous, statistically solid analysis using the many patient samples available.
2) In the previous round this reviewer commented on another conclusion that appears in the abstract, “LOLA reduced the release of NH3; beneficially modulated the expression of genes related to fatty acid import/transport (cd36, cpt1), synthesis (fasn, 28 scd1, ACC1), and regulation (srbf1); reduced cellular ATP and acetyl-CoA; and favorably modulated the expression of master regulators/genes of energy balance/mitochondrial biogenesis (AMPK-α,PGC1α).”However, in this V2 version, it still remains that, while abundance of FASN mRNA transcript is decreased by LOLA, FASN protein is not. That questions the ability of LOLA to change significantly, at the protein level, “gene expressions related to fatty acid import, synthesis, and regulation”. It is noted that the analysis is done only at the mRNA level for all other parameters in Fig. 4. In respect to modulation of genes of energy balance, such as that of activity of AMPK, here monitored by its level pf phosphorylation in Fig. 5C, it does not appear as significantly changed by LOLA, again raising questions on the conclusions stated in the abstract.
Reviewer 2 Report
Comments and Suggestions for Authors
The authors now plan followup studies to assess specificity of amino acid effects that they demonstrated with LOLA. That meets my criticisms and certainly they have performed an impressive analysis of LOLA effects